# Robust Trajectory Prediction against Adversarial Attacks

Yulong Cao[*1,2], Danfei Xu[2,3], Xinshuo Weng[2], Z. Morley Mao[1], Anima Anandkumar[2,4], Chaowei Xiao[2,5], and Marco Pavone[2,6]

[1]University of Michigan
[2]NVIDIA
[3]Georgia Institute of Technology
[4]California Institute of Technology
[5]Arizona State University
[6]Stanford University

**Abstract:** Trajectory prediction using deep neural networks (DNNs) is an essential component of autonomous driving (AD) systems. However, these methods are vulnerable to adversarial attacks, leading to serious consequences such as collisions. In this work, we identify two key ingredients to defend trajectory prediction models against adversarial attacks including (1) designing effective adversarial training methods and (2) adding domain-specific data augmentation to mitigate the performance degradation on clean data. We demonstrate that our method is able to improve the performance by 46% on adversarial data and at the cost of only 3% performance degradation on clean data, compared to the model trained with clean data. Additionally, compared to existing robust methods, our method can improve performance by 21% on adversarial examples and 9% on clean data. Our robust model is evaluated with a planner to study its downstream impacts. We demonstrate that our model can significantly reduce the severe accident rates (e.g., collisions and off-road driving)[1].

## 1 Introduction

Trajectory prediction is a critical component of modern autonomous driving (AD) systems. It allows an AD system to anticipate the future behaviors of other nearby road participants and plan its actions accordingly. Recent trajectory prediction models built on Deep Neural Networks (DNN) have demonstrated state-of-the-art performance on large-scale benchmarks [1–7], showing a promising path towards learning-based trajectory prediction for AD systems. As trajectory prediction plays an important role in AD systems, accurate predictions are required for making safe driving decisions. It is crucial to understand how unknown scenarios will affect trajectory predictions and then bolster the robustness of such trajectory predictions in return.

To achieve this goal, adversarial attacks [8–10] are often used as a proxy to measure the worst-case performance of the model when facing unseen scenarios. Similarly, we use a standard adversarial attack setup [11] for trajectory predictions. As illustrated in Fig. 1, an adversarial agent (red vehicle) aims to cause a traffic accident. It drives along a carefully designed trajectory (i.e., adv history) to influence the trajectory prediction model of the Autonomous Vehicle (green vehicle). Such an adversary can critically compromise the predicted trajectories of all other agents by altering its route in inconspicuous ways. By fooling the trajectory prediction models, it can further affect downstream planning of the AV systems and cause serious consequences. Using the adversarial attack as the proxy, this work aims to develop effective techniques to bolster the robustness of trajectory prediction models against adversarial attacks and improve the AD's safety under uncertain scenarios.

---

[*]Work done during an internship at NVIDIA
[1]Our project website is at https://robustav.github.io/RobustTraj

6th Conference on Robot Learning (CoRL 2022), Auckland, New Zealand.

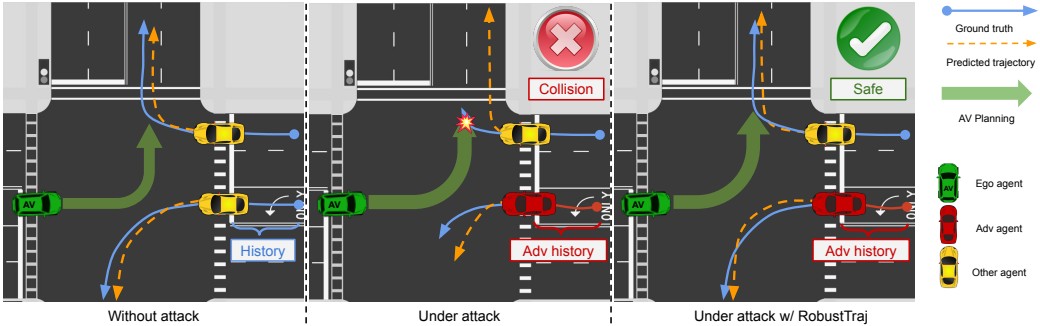

Figure 1: Overview of *RobustTraj* preventing Autonomous Vehicle (AV) from collisions when its trajectory prediction model is under adversarial attacks. When the trajectory prediction model is under attack, the AV predicts the wrong future trajectory of the other agent turning right (yellow vehicle). This results in AV speeding up instead of slowing down, and eventually colliding into the other vehicle.

At the same time, adversarial robustness for machine learning is a widely-studied area, but most works focus on classification tasks [12–23]. Among the proposed techniques, adversarial training [8] remains the most effective and widely used method to defend classifiers against adversarial attacks. The general strategy of adversarial training is to solve a min-max game by generating adversarial examples for a model at each training step and then optimizing the model to make correct predictions for these samples. However, directly applying adversarial training to trajectory prediction presents a number of critical technical challenges.

First, most trajectory prediction methods employ probabilistic generative models to cope with the uncertainty in motion forecasting [2–7]. As we will show in this paper, the stochastic components of these models (e.g., posterior sampling in VAEs) can obfuscate the gradients that guide the adversarial generation, making naïve adversarial training methods ineffective. Second, adversarial training on trajectory prediction task aims to model joint data distribution of future trajectories and adversarial past trajectories. However, the co-evolution of the adversarial sample distribution and the prediction model during the training process makes the joint distribution hard to model and often destabilizes the adversarial training. Finally, prior work [23] shows that adversarial training often leads to degraded performance on clean (unperturbed) data, while retaining good performance in benign cases is crucial due to the critical role of trajectory prediction for AVs. Hence, an effective adversarial training method must carefully balance the benign and the adversarial performance of a model.

**Our approach.** We propose an adversarial training framework for trajectory predictions named *RobustTraj*, by addressing the aforementioned challenges. First, to address the issue of an obfuscated gradient in adversarial generation due to stochastic components, we devise a *deterministic attack* that creates a deterministic gradient path within a probabilistic model to generate adversarial samples. Second, to address the challenge of an unstable training process due to shift in adversarial distributions, we introduce a hybrid objective that interleaves the adversarial training and learning from clean data to anchor the model output on stable clean data distribution. Finally, to achieve balanced performances on both adversarial and clean data, we introduce a domain-specific data augmentation technique for trajectory prediction via a dynamic model. This data augmentation technique generates diverse, realistic, and dynamically-feasible samples for training and achieves a better performance trade-off on clean and adversarial data.

We empirically show that *RobustTraj* can effectively defend two different types of probabilistic trajectory prediction models [4, 7] against adversarial attacks, while incurring minimal performance degradation on clean data. For instance, *RobustTraj* can increase the adversarial performance of AgentFormer [4], a state-of-the-art trajectory prediction model, by 46% at the cost of 3% performance drop on clean data. To further show impacts of our method on the AD stack, we plug our robust trajectory prediction model into a planner and demonstrate that our model reduces serious accidents rates (e.g., collisions and off-road driving) under attacks by 100%, compared to the standard non-robust model trained using only clean data.

## 2 Related Work

**Adversarial attacks and defenses on trajectory prediction.** A recent work began to study the adversarial robustness of trajectory prediction models [11]. Zhang et al. [11] demonstrated that perturbing agents' observed trajectory can adversarially impact the prediction accuracy of a DNN-based trajectory forecasting model. To mitigate the issue, Zhang et al. [11] proposed several defense methods such as data augmentation and trajectory smoothing. However, these methods are less effective when facing adaptive attacks [24]. In our work, we propose to use adversarial training which provides the general adversarial robustness that can resist adaptive attacks.

**Adversarial scenario generation.** A few recent studies work on generating adversarial traffic scenarios such that the autonomous driving systems fail to make safe driving decisions [25, 26]. However, generating realistic traffic scenarios is challenging and the generated adversarial scenarios can be unrealistic and violate traffic rules by directly optimizing the latent vectors of the traffic model Rempe et al. [26]. In this work, we consider defending against realistic adversarial scenarios grounded on the scenarios from a dataset. Wang et al. [25] perturb the raw input data to mislead the full stack AV system. However, in this work, our primary goal is to study and improve the robustness of trajectory prediction models. To obtain salient and unambiguous insights, we minimize the conflating factors in our analysis without considering the perception model.

**Adversarial training.** A variety of adversarial training methods have been proposed to defend DNN-based models against adversarial attacks [8, 19, 12–23]. The most common strategy is to design a min-max game with the inner maximization process and outer minimization process. The inner maximization process generates adversarial examples that maximize an adversarial objective (e.g., make wrong prediction). The outer minimization process then updates the model parameters to minimize the error on the adversarial examples. Several recent works also propose to mix clean data and adversarial examples for improving robustness [27, 28] and performance on clean data [18]. Although there exists a large body of literature in studying adversarial robustness for machine learning, most focus on the problem of discriminative model (e.g., object recognition), leaving other problem domains (e.g., conditional generative models) largely unexplored. In this work, we develop a novel adversarial training method for trajectory prediction models, where most state-of-the-art trajectory prediction models are generative and probabilistic, by addressing a number of critical technical challenges.

## 3 Preliminaries and Formulation

**Trajectory prediction.** The goal is to predict future trajectory distribution $\mathcal{P}_\theta(Y|X)$ of $N$ agents conditioned on their $H$ history time states $\mathbf{X} = \left(\mathbf{X}^{-H+1}, \ldots, \mathbf{X}^0\right)$, and other environment context such as maps [2] to predict $T$ time-step future trajectories $\mathbf{Y} = \left(\mathbf{Y}^1, \ldots, \mathbf{Y}^T\right)$. For observed time steps $t \leq 0$, we denote the agent states as $\mathbf{X}^t = (x_1^t, \ldots, x_i^t, \ldots, x_N^t)$, where $x_i^t$ is the state of agent $i$ at the time step $t$. Similarly, $\mathbf{Y}^t = (y_1^t, \ldots, y_N^t)$ denotes the states of $N$ agents at a future time step $t$ ($t > 0$). We denote the ground truth and the predicted future trajectories as $\mathbf{Y}$ and $\hat{\mathbf{Y}}$, respectively. We denote the history information encoded by a function $f$ as the decision context $\mathbf{C} = f(\mathbf{X})$.

**Probabilistic trajectory prediction models.** In this work, we focus on defending generative, probabilistic trajectory prediction models, as they have demonstrated superior performance in modeling uncertainty in predicting future motions [2–7]. We consider the two most popular types of generative models: conditional variational encoders (CVAEs) and conditional GANs (cGANs), both can be viewed as latent variable models. We define latent variables $\mathbf{Z} = \{z_1, \ldots, z_i, \ldots, z_N\}$ where $z_i$ represents the latent variable of the agent $i$. CVAE formulates the generative problem as: $p_\theta(\mathbf{Y}|\mathbf{X}) = \int p_\theta(\mathbf{Y}|\mathbf{X}, \mathbf{Z}) \cdot p_\theta(\mathbf{Z}|\mathbf{X}) d\mathbf{Z}$, where $p_\theta(\mathbf{Z}|\mathbf{X})$ is a conditional Gaussian prior ($\mathcal{N}(p_\theta^\mu(\mathbf{Z}|\mathbf{X}), p_\theta^\sigma(\mathbf{Z}|\mathbf{X}))$) with mean $p_\theta^\mu(\mathbf{Z}|\mathbf{X})$ and standard deviation $p_\theta^\sigma(\mathbf{Z}|\mathbf{X})$; $p_\theta(\mathbf{Y}|\mathbf{X}, \mathbf{Z})$ is a conditional likelihood model. The model is usually trained through optimizing a negative evidence lower objective [4]:

$$\begin{aligned}
\mathcal{L}_{\text{total}} &= \mathcal{L}_{\text{elbo}} + \mathcal{L}_{\text{diversity}} \\
&= -\mathbb{E}_{q_\phi(\mathbf{Z}|\mathbf{Y},\mathbf{X})}[\log p_\theta(\mathbf{Y}|\mathbf{Z}, \mathbf{X})] + \text{KL}(q_\phi(\mathbf{Z}|\mathbf{Y}, \mathbf{X}) \| p_\theta(\mathbf{Z}|\mathbf{X})) + \min_k \| \hat{\mathbf{Y}}^{(k)} - \mathbf{Y} \|^2,
\end{aligned} \quad (1)$$

---

[2] For simplicity, we ignore contextual information.

where $q_\phi(\mathbf{Z}|\mathbf{Y}, \mathbf{X})$ is an approximate posterior parameterized by $\phi$, $p_\theta(\mathbf{Z}|\mathbf{X})$ is a conditional Gaussian prior parameterized by $\theta$, and $p_\theta(\mathbf{Y}|\mathbf{Z}, \mathbf{X})$ is a conditional likelihood modeling future trajectory $\mathbf{Y}$ via the latent codes $\mathbf{Z}$ and past trajectory $\mathbf{X}$. Additionally, $\mathcal{L}_{\text{diversity}} = \min_k \| \hat{\mathbf{Y}}^{(k)} - \mathbf{Y} \|^2$ is a diversity loss, which encourages the network to produce diverse samples. Given each past trajectory $X$, the model generates $K$ sets of latent codes $\{\mathbf{Z}^{(1)}, \cdots, \mathbf{Z}^{(k)}, \cdots, \mathbf{Z}^{(K)}\}$ from the conditional Gaussian prior $\mathcal{N}(p_\theta^\mu(\mathbf{Z}|\mathbf{X}), p_\theta^\sigma(\mathbf{Z}|\mathbf{X}))$, where $\mathbf{Z}^{(k)} = \{z_1^k, \cdots, z_n^k\}$, resulting in $K$ future trajectories $\hat{\mathbf{Y}}^{(k)}$.

Similarly, in a conditional Generative Adversarial Net (cGAN)-based model (e.g., Social-GAN [1]), it uses a loss function as follows:

$$\begin{aligned}
\mathcal{L}_{\text{total}} &= \mathcal{L}_{\text{gan}} + \mathcal{L}_{\text{diversity}} \\
&= \mathbb{E}_{\mathbf{Y} \sim p_{\text{data}}}[\log D_\theta(\mathbf{Y}|\mathbf{X})] + \mathbb{E}_{\mathbf{Z} \sim p_Z}[\log(1 - D_\theta(G_\phi(\mathbf{Y}|\mathbf{X}, \mathbf{Z})))] + \min_k \| \hat{\mathbf{Y}}^{(k)} - \mathbf{Y} \|^2,
\end{aligned} \quad (2)$$

where $G$ represents the generator and $D$ represents the discriminator. $\hat{\mathbf{Y}}^{(k)} = G(\mathbf{Y}|\mathbf{X}, \mathbf{Z}^{(k)})$ is one of the predicted trajectories in which $\mathbf{Z}^{(k)}$ is randomly sampled from $\mathcal{N}(0, 1)$. During the training, $\mathcal{L}_{\text{gan}}$ is maximized to train $D$ and $\mathcal{L}_{\text{total}}$ is minimized to train $G$.

**Threat model.** We follow the setup in prior work [11] and adopt an idealized threat model, where the adversary alters its observed history $\mathbf{X}$ by adding a perturbation $\delta$ bounded by the adversarial set $\mathbb{S}_p^\epsilon = \{\delta| \| \delta \|_p \leq \epsilon\}$, where $\epsilon$ is the perturbation bound. The perturbation aims to mislead the prediction $\hat{\mathbf{Y}}$. A naïve adversarial attack is to find the perturbation through an adversarial objective $\delta = \arg\max_{\delta \in \mathbb{S}}\{\min_{k \in \{1, \cdots, K\}} \| p_\theta(\mathbf{Y}|\mathbf{X} + \delta, \mathbf{Z}^{(k)}) - \mathbf{Y} \|^2\}$, where $p_\theta(\mathbf{Y}|\mathbf{X} + \delta, \mathbf{Z}^{(k)})$ is the predicted trajectory conditioned on the random variable $\mathbf{Z}^{(k)}$ and adversarial history trajectory $\mathbf{X} + \delta$.

**Naïve adversarial training.** Adversarial training formulates a min-max game with an inner maximization process that optimizes the perturbation $\delta$ to generate adversarial examples for misleading the model at each training iteration, and an outer minimization process that optimizes the model parameters to make correct predictions for these examples. We follow the standard adversarial training formulation [8]:

$$\min_{\theta, \phi} \max_{\delta \in \mathbb{S}} \quad \mathcal{L}_{\text{total}}(\mathbf{X} + \delta, \mathbf{Y}). \quad (3)$$

# 4  *RobustTraj*: Robust Trajectory Prediction

As stated earlier, applying adversarial training for trajectory prediction presents three critical challenges: (1) gradient obfuscation due to model stochasticity, (2) unstable learning due to changing adversarial distribution, and (3) performance loss in the benign situation. In this section, we describe each challenge in more detail and present the corresponding solutions in our *RobustTraj* method.

**Improve adversarial generation with *Deterministic Attack*.** Since trajectory prediction is inherently uncertain and there is no single correct answer, probabilistic generative models are usually used to cope with the stochastic nature of the trajectory prediction task. Such stochasticity will obfuscate the gradients that are used to generate effective adversarial examples in the inner maximization process of adversarial training. The naïve attack mentioned in section 3 is a straightforward way to achieve this goal. However, this optimization involves a stochastic sampling process $\mathbf{Z}^{(k)} \sim \mathcal{N}(p_\theta^\mu(\mathbf{Z}|\mathbf{X}), p_\theta^\sigma(\mathbf{Z}|\mathbf{X}))$. Such a stochastic process will obfuscate the gradients for finding the optimal adversarial perturbation $\delta$, making the outer minimization (robust training) less effective. In order to sidestep such stochasticity, we propose the *deterministic attack* that creates a deterministic gradient path within the model to generate the adversarial perturbation. $\hat{\mathbf{Z}}$. Specifically, we use a *deterministic latent code* by replacing the sampling process $\mathbf{Z}^{(k)} \sim \mathcal{N}(p_\theta^\mu(\mathbf{Z}|\mathbf{X}), p_\theta^\sigma(\mathbf{Z}|\mathbf{X}))$, with the maximum-likelihood sample (here, i.e $\hat{\mathbf{Z}} = p_\theta^\mu(\mathbf{Z}|\mathbf{X})$). The objective for generating the adversarial perturbation is thus:

$$\delta = \arg\max_{\delta \in \mathbb{S}} \mathcal{L}_{\text{adv}}(\mathbf{X} + \delta, \mathbf{Y}) = \arg\max_{\delta \in \mathbb{S}} \| p_\theta(\mathbf{Y}|\hat{\mathbf{Z}}, \mathbf{X} + \delta) - \mathbf{Y} \|^2, \text{ where } \hat{\mathbf{Z}} = p_\theta^\mu(\mathbf{Z}|\mathbf{X} + \delta). \quad (4)$$

We empirically show that gradients from this deterministic gradient path can effectively guide the generation of adversarial examples. We name our attack as *Deterministic Attack*.

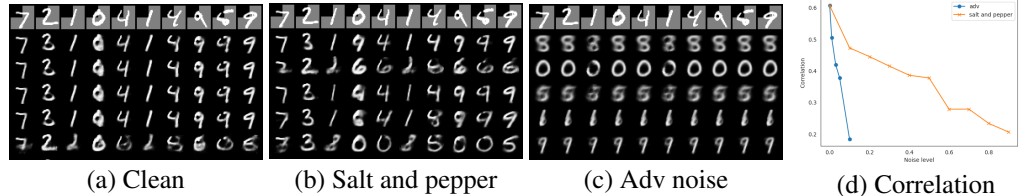

| (a) Clean | (b) Salt and pepper | (c) Adv noise | (d) Correlation |

Figure 2: Visualizations of the CVAE models trained with clean (a) data, *Salt and pepper* noise (b), and adversarial perturbations (c); Quantitative results of the correlation between the label of the generated images and conditioned images at different noise levels (d).

**Stabilize adversarial training with bounded noise and hybrid objective.** During the adversarial training process, the distribution of the perturbed input $\mathbf{X} + \delta$ coevolves with the training process as $\delta$ is calculated via an inner maximization process at each training iteration. Although $\delta$ is bounded by the adversarial set $\mathbb{S}_p^\epsilon$, the resulting latent condition variable $\mathbf{C} = f(\mathbf{X} + \delta)$ can be arbitrarily noisy since the Lipschitz constant of neural network layers ($f$) is not bounded during training (See *Lemma 1.* in Appendix A). Since $\mathbf{C} = f(\mathbf{X} + \delta)$ is noisy, it is a less informative signal compared to the deterministic signal $X$. Thus, modeling $p_\theta(\mathbf{Y}|\mathbf{X} + \delta, \mathbf{Z})$ becomes substantially harder. In an extreme case that $\mathbf{C} = f(\mathbf{X} + \delta)$ is super noisy and contains no information, the training process can degenerate to model $p_\theta(\mathbf{Y}|\mathbf{Z})$, resulting in the undesirable worse performance on the clean data.

To further validate the above hypothesis that it is hard to model $p_\theta(\mathbf{Y}|\mathbf{X} + \delta, \mathbf{Z})$ with a changing data distribution of $\mathbf{X} + \delta$, we conduct an additional experiment. For simplicity, we use MNIST [29] as the dataset. As shown in Fig. 2, we divide each digit image into four quadrants. We take the top-left quadrant as the condition $\mathbf{X}$ and the remaining quadrants as the output $\mathbf{Y}$. We train a CVAE ($p_\theta(\mathbf{Y}|\mathbf{X} + \delta, \mathbf{Z})$) to model $\mathbf{Y}$ by using clean data ($\mathbf{X}$) or noisy data ($\mathbf{X} + \delta$), where $\delta$ represents salt and pepper noise [30] or adversarial noise [8], resulting in Fig. 2 (a), (b), (c) respectively. The top-left region of each image in the first row is the conditional variables $\mathbf{X}$. The rest of rows are the generated images with different $\mathbf{Z}$. Each column in the same row uses the same $\mathbf{Z}$. We observe that the model trained on clean data successfully captures the conditional distribution (i.e., the generated image highly depends on $\mathbf{X}$) while the model trained with adversarial noise degenerates and ignores the condition (i.e., each row generates images of the same digit). This result shows that the conditional generative model fails to learn from $\mathbf{X}$. To provide a quantitative analysis, we measure the correlation between the label of the generated images and the label of their conditioned image quadrants, resulting in Fig 2 (d). More details on how to calculate the correlation are in the Appendix A. We observe that the correlation drops as the noise level increases for both adversarial nose and *salt and pepper* noise. Adversarial noise is more effective to degenerate the conditional generative model. Therefore, we conclude that (1) the noises in the conditional data lead to degenerated conditional generative model (i.e., from CVAE to VAE); (2) the level of degeneration depends on the noise levels.

Based on the analysis result, to better learn a robust trajectory prediction model, we need to bound $|f(\mathbf{X} + \delta) - f(\mathbf{X})|$ to reduce the noise level. Hence, we propose the following regularization loss $\mathcal{L}_{\text{reg}}$:

$$\mathcal{L}_{\text{reg}} = \mathrm{d}(f(\mathbf{X} + \delta), f(\mathbf{X})), \tag{5}$$

where $\mathrm{d}$ is a distance function (e.g., we use $L_2$ norm as the distance metric).

In addition, because the clean data has a fixed distribution, simultaneously learning from the clean data during the adversarial training process anchors the conditional distribution on a stable clean data distribution. Specifically, we propose the following hybrid objective:

$$\mathcal{L}_{\text{clean}}(\mathbf{X}, \mathbf{Y}) = \mathcal{L}_{\text{total}}(\mathbf{X}, \mathbf{Y}), \tag{6}$$

where $\mathcal{L}_{\text{total}}$ could be the loss in Eq. 1 for CVAE-based model or Eq. 2 for cGAN-based model.

**Protect benign performance using data augmentation.** Adversarial training often leads to performance degradation on clean data [23]. However, trajectory prediction is a critical component for safety-critical AD systems and its performance degradation can result in severe consequences (e.g., collisions). Thus, it is important to balance the model performance on the clean and adversarial data when designing adversarial training algorithms.

To further improve the performance on clean and adversarial data, we need to address the overfitting problem of the min-max adversarial training [31]. Data augmentation is shown to be effective

in addressing the problem in the image classification domain [32]. However, data augmentation in trajectory prediction is rarely studied and non-trivial. To design an effective augmentation algorithm, Rebuffi et al. [32] argues that the most important criterion is that the augmented data should be realistic and diverse. Thus, we design a dynamic-model based data augmentation strategy $\mathbb{A}$ shown in the Appendix A. By using the augmentation, we can generate diverse, realistic multi-agent trajectories for each scene and construct $\mathbb{D}_{\text{aug}}$.

***RobustTraj.*** In summary, our adversarial training strategy for trajectory prediction models is formulated as follows:

$$
\begin{aligned}
\delta &= \arg\max_{\delta \in \mathbb{S}} \mathcal{L}_{\text{adv}}(\mathbf{X} + \delta, \mathbf{Y}), \quad \text{where}\{\mathbf{X}, \mathbf{Y}\} \in \mathbb{D} \cup \mathbb{D}_{\text{aug}} \\
\theta, \phi &= \arg\min_{\theta, \phi} \mathcal{L}_{\text{total}}(\mathbf{X} + \delta, \mathbf{Y}) + \mathcal{L}_{\text{clean}}(\mathbf{X}, \mathbf{Y}) + \beta \cdot \mathcal{L}_{\text{reg}}(\mathbf{X}, \mathbf{X} + \delta),
\end{aligned}
\tag{7}
$$

where $\mathbb{D}, \mathbb{D}_{\text{aug}}$ are the training data and augmented data; $\mathcal{L}_{\text{adv}}$ is adversarial loss to generate effective adversarial examples in Eq. 4; $\mathcal{L}_{\text{total}}$ is the loss in Eq. 1 or Eq. 2 to train a robust model against adversarial examples; $\mathcal{L}_{\text{reg}}$ and $\mathcal{L}_{\text{clean}}$ are loss shown in Eq. 5 and Eq. 6 to provide a stable signal for training. $\beta$ is a hyper-parameter for adjusting the regularization.

## 5 Experiments and Results

### 5.1 Experimental setup

**Dataset and models.** We follow the setting in prior work [4, 3] and use the nuScenes dataset [33] for evaluation. For the trajectory prediction models, we select the representative conditional generative models based on CVAE (AgentFormer [4]) and cGAN (Social-GAN [1]). AgentFormer is a state-of-the-art model based on CVAE and Social-GAN is a classic model based on cGAN. We report the final results for all three models: AgentFormer (AF), mini-AgentFormer (mini-AF) and Social-GAN. More details are shown in the Appendix B.

**Training details and hyperparameter choices.** For the adversarial training, we choose a 2-step Projected Gradient Descent (PGD) attack for the inner maximization and choose $\beta = 0.1$. We train 50 epochs and 100 epochs for AgentFormer and Social-GAN respectively. For other hyperparameters during training, we follow the original settings for AgentFormer and Social-GAN. The details for choosing these hyperparameters can be found in the Appendix B. All experiments are done on the NVIDIA V100 GPU [34]. We consider various baselines, including naïve adversarial training (naïve AT) and four defenses proposed by Zhang *et al.* [11]: data augmentation with adversarial examples (*DA*), *train-time smoothing*, *test-time smoothing*, *DA + train-time smoothing* and *detection + test-time smoothing*.

**Attack and evaluation metrics.** For the adversarial attack, we choose a 20-step PGD attack (an ablation study on step convergence can be found in the Appendix B). Without loss of generality, we use $L_\infty$ as the attack threat model so that $\mathbb{S} = \{\delta | \parallel \delta \parallel_\infty \leq \epsilon\}$. We select $\epsilon = \{0.5, 1.0\}$-meter, where the 1-meter deviation is the maximum change for a standard car without shifting to another lane [11]. We use four standard evaluation metrics for the nuscenes prediction challenge [33]: average displacement error (ADE), final displacement error (FDE), off road rates (ORR), and miss rate (MR). We evaluate the model's performance on both clean and adversarial data. For convenience, we use *ADE*, *FDE*, *ORR*, *MR* to represent the performance on the clean data and *Robust ADE*, *Robust FDE*, *Robust ORR*, *Robust MR* to represent the performance under attacks. We compute these metrics with the best of five predicted trajectory samples, i.e., $K = 5$.

### 5.2 Main results

Here, we present our main results of *RobustTraj*. We compare it with the baselines including model trained with clean data (*Clean*) and naïve adversarial training (*Naïve AT*), and existing defense methods for trajectory prediction [11]. The results have been shown in Table 1.

We observe that our method achieves the best robustness and maintains good clean performance for most cases. For instance, with $\epsilon = 0.5$ attack on AgentFormer model, our method is able to reduce 46% prediction errors ($\frac{5.09-2.73}{5.09}$) under the attack at a cost of 2.6% ($\frac{1.91-1.86}{1.86}$) clean performance degradation on ADE, compared to the model trained with clean data at $\epsilon = 0.5$. Compared to the

Table 1: ADE and Robust ADE on different defense methods and models. The 1-st and 2-nd lowest errors are colored.

| Model | mini-AF | | | | AF | | | | SGAN | | | |
|---|---|---|---|---|---|---|---|---|---|---|---|---|
| Method | ADE | | Robust ADE | | ADE | | Robust ADE | | ADE | | Robust ADE | |
| | 0.5 | 1.0 | 0.5 | 1.0 | 0.5 | 1.0 | 0.5 | 1.0 | 0.5 | 1.0 | 0.5 | 1.0 |
| Clean | **2.05** | **2.05** | 6.86 | 11.53 | **1.86** | **1.86** | 5.09 | 8.57 | **4.80** | **4.80** | 10.52 | 20.15 |
| *Naïve AT [8]* | 2.75 | 2.78 | 5.44 | 9.20 | 2.52 | 2.56 | **3.81** | 6.81 | 6.43 | 6.55 | **8.34** | **14.63** |
| *DA [11]* | 2.31 | 2.32 | 5.54 | 9.32 | 2.10 | 2.08 | 4.35 | 7.22 | 5.41 | 5.40 | 8.85 | 17.25 |
| *Train-time Smoothing [11]* | 3.14 | 3.07 | 5.67 | 9.31 | 2.11 | 2.13 | 4.19 | 6.79 | 5.50 | 5.47 | 8.74 | 16.51 |
| *Test-time Smoothing [11]* | 2.97 | 3.07 | **4.96** | **8.50** | 2.40 | 2.41 | 4.43 | 7.44 | 6.16 | 6.17 | 9.05 | 17.42 |
| *DA + Train-time Smoothing [11]* | 2.41 | 2.39 | 5.48 | 9.00 | 2.17 | 2.13 | 4.14 | **6.62** | 5.63 | 5.61 | 8.60 | 16.14 |
| *Detection + Test Smoothing [11]* | 2.31 | 2.28 | 5.91 | 9.85 | 2.08 | 2.03 | 4.45 | 7.59 | 5.35 | 5.37 | 9.28 | 17.39 |
| ***RobustTraj*** | **2.14** | **2.11** | **3.69** | **3.82** | **1.91** | **1.95** | **2.73** | **2.86** | **4.95** | **5.07** | **5.20** | **6.94** |

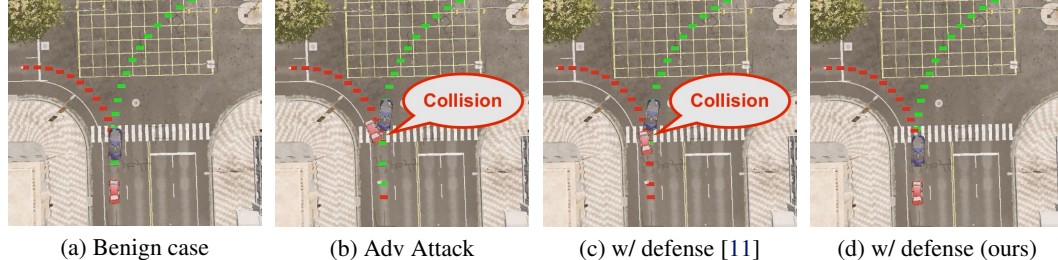

(a) Benign case    (b) Adv Attack    (c) w/ defense [11]    (d) w/ defense (ours)

Figure 3: Impacts to a MPC-basd downstream planner. (a) is under the benign case while (b), (c) and (d) are under the adversarial attacks. The blue car and the red car represent the AV and the adversarial agent respectively.

existing methods, our method also significantly outperforms in terms of the robustness. For instance, with $\epsilon = 1.0$ attack on AgentFormer model, our method achieves 45% better robustness with 9% better clean performance on ADE compared to the best results from existing methods [11].

**Impacts to downstream planners.** To further study the downstream impact of our robust trajectory model in the AD stack, we plug it into a planner. We select a MPC-based planner and evaluate the collision rates under the attack. To perform the attack on a closed-loop planner, we conduct attacks on a sequence of frames with the expectation over transformation (EOT) [35] method. We follow the setting from Zhang et al. [11] and choose $\epsilon = 1$. We choose AgentFormer model since it has the most competitive performance. As a result, we observe that, while AgentFormer model trained on clean data leads to 10 collision cases under attack, the robust trained model with the proposed *RobustTraj* is able to avoid all the collisions. As shown in Fig. 3, we demonstrate that the proposed *RobustTraj* is able to avoid the collisions (Fig. 3 (d)) while the *DA + Train-time Smoothing* method proposed by Zhang et al. [11] is not (Fig. 3 (c)).

## 5.3 Component analysis

In this section, we analyze the effectiveness of the three components. We use the mini-AgentFormer model since it has competitive performance and is lightweight for a fast adversarial training process.

**Effectiveness of the *Deterministic Attack*.** To demonstrate the importance of the *Deterministic Attack*, we compare it with competitive alternatives, *Latent Attack* and *Context Attack*, which also construct the deterministic path. However, they only attack a partial model as opposed to our end-to-end full model attack. More details about these attacks are in the Appendix A. We evaluate their attack effectiveness by attacking a normally trained trajectory prediction model (without robust training). In Fig. 4, we demon-

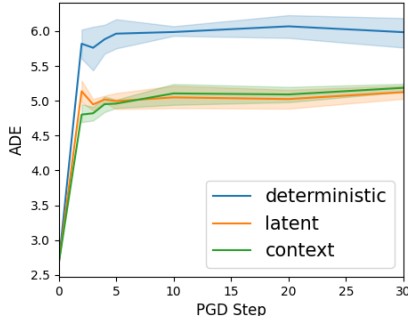

Figure 4: Peformance of different attacks in mini-AgentFormer.

strate that *Deterministic Attack* is the most effective attack among all. Additionally, we embed them into the whole adversarial training pipeline and evaluate the adversarial robustness. The results are shown in Table 2. We observe that the model trained with the *Deterministic Attack* achieves the best robustness in terms of ADE. More results with other metrics and the other $\epsilon$ are in the Appendix B.

**Effect of additional loss functions.** We evaluate the performance of the models trained with additional loss terms: $\mathcal{L}_{\text{clean}}$ and $\mathcal{L}_{\text{reg}}$. In Table 2, we can see that the regularization term $\mathcal{L}_{\text{reg}}$ improves robustness of the models and achieves better clean performance. It shows that the regularization of the introduced noises on conditional variables help the model to stabilize the training procedure. By adding the clean loss $\mathcal{L}_{\text{clean}}$, we observe that both the robustness and clean performance are improved further, which means the benign data indeed anchors the model output on clean data distribution and provides a stable signal for the better robust training for generative models.

Table 2: ADE and robust ADE for different methods on mini-AgentFormer. The lowest error is in bold.

| Method | ADE | | Robust ADE | |
|---|---|---|---|---|
| | 0.5 | 1.0 | 0.5 | 1.0 |
| Clean | 2.05 | 2.05 | 6.86 | 11.53 |
| *Latent Attack* | 2.55 | 2.70 | 4.10 | 4.71 |
| *Context Attack* | **2.47** | 2.59 | 3.94 | 4.78 |
| *Deterministic Attack* | 2.61 | **2.55** | **3.88** | **4.35** |
| *Deterministic Attack* | | | | |
| + $\mathcal{L}_{\text{reg}}$ | 2.29 | 2.31 | 3.76 | 4.28 |
| + $\mathcal{L}_{\text{clean}}$ + $\mathcal{L}_{\text{reg}}$ | 2.23 | 2.19 | 3.71 | 3.83 |
| + $\mathcal{L}_{\text{clean}}$ + $\mathcal{L}_{\text{reg}}$ + Aug | **2.14** | **2.11** | **3.69** | **3.82** |

**Effect of domain-specific augmentation.** To demonstrate the effectiveness of the domain-specific augmentation, We also combine it with all of the above components to validate its effect. The results are shown in Table 2. We observe that it achieves a better performance on clean and adversarial data.

# 6 Limitations

In this work, we identified the challenges of applying adversarial training on trajectory prediction models based on probabilistic generative models since they could cope with the natural uncertainty of motion forecasting. Though the probabilistic generative model is the main-stream architecture for the trajectory prediction task, there are other architectures (e.g., LSTM [1, 36], flow-based method [5, 6] and RL-based method [37]) for generating multi-modal predictions. Additionally, we only study the adversarial set with the threat model of $\mathcal{L}_\infty$ perturbation on trajectories instead of other types of threat models (e.g., optimization on the latent space [26], perturbation on raw sensor data [25]). Moreover, the primary goal of this paper is to study and improve the robustness of trajectory prediction models. To obtain salient and unambiguous insights, we minimize the conflating factors in our analysis without considering the perception model in our pipeline. We leave these as future work for building robust trajectory prediction models.

# 7 Conclusion

In this paper, we aim to study how to train robust generative trajectory prediction models against adversarial attacks, which is seldom explored in the literature. To achieve this goal, we first identify three key challenges in designing an adversarial training framework to train robust trajectory prediction models. To address them, we propose an adversarial training framework with three main components, including (1) a *deterministic attack* for the inner maximization process of the adversarial training, (2) additional regularization terms for stable outer minimization of adversarial training, and (3) a domain-specific augmentation strategy to achieve a better performance trade-off on clean and adversarial data. To show the generality of our method, we apply our approach to two trajectory prediction models, including (1) a CVAE-based model, AgentFormer, and (2) a cGAN-based model, Social-GAN. Our extensive experiments show our method could significantly improve the robustness with a slight performance degradation on the clean data, compared to the existing techniques and dramatically reduce the severe collision rates when plugged into the AD stack with a planner. We hope our work can shed light on developing robust trajectory prediction systems for AD.

## Acknowledgement

This work was supported by NSF under the National AI Institute for Edge Computing Leveraging Next Generation Wireless Networks, Grant # 2112562, as well as NSF grant CNS-1930041 and CMMI-2038215.

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
