# OpenReview forum: "Robust Trajectory Prediction against Adversarial Attacks"
_robot-learning.org/CoRL/2022/Conference — CoRL 2022 Oral_

### Official Review · Reviewer_xfRa · 2022-07-27

**Originality:** Very Good
**Technical Quality:** Very Good
**Clarity Of Presentation:** Very Good
**Impact:** 3

**Recommendation:**

Weak Accept: I recommend accepting the paper, but will not argue for my recommendation if the majority of other reviewers have a different opinion.

**Summary:**

This work aims to study how to train robust generative trajectory prediction models against adversarial attacks.
The authors list 3 major challenges for this task and correspondingly design 3 components for RobustTraj:  (1) deterministic attack, (2)training with bounded noise and hybrid objective  (3) data augmentation.
RobustTraj is applied on two trajectory prediction models. Results on nuScenes shows that the proposed method could significantly improve the robustness with small degradation on the clean data.

**Issues:**

If the authors can address my concern in the weakness. I may be wrong.
It would be more strong, if it can be tested in ArgoVerse motion forecasting dataset too.

**Quality Of The Limitations Section:**

Additional details required

**Reviewer Expertise:**

3: The reviewer is fairly confident that the evaluation is correct

**Robotics Focus:**

Highly relevant to robotics but no hardware experiments

**Strengths And Weaknesses:**

## strengths
The authors clearly describe the challenges for adversarial training on trajectory prediction, and propose several reasonable solutions to address these issues.

According to the experimental results, the proposed method could significantly improve the robustness with only small degradation on the clean data. I hope this work can help build more robust trajectory prediction work

The authors  also plug it into a planner and demonstrate that the robust trained model is able to avoid all the collisions.

## Weakness
Lack of generalizability? The proposed adversarial training method may not be feasible/necessary for trajectory prediction methods like LSTM which directly predict future trajectory without sampling.

**Summary Of Recommendation:**

This paper is generally convincing with thorough experiments. The improved robustness shown in the supplementary material looks attractive.

---

### Official Review · Reviewer_Bth3 · 2022-07-31

**Originality:** Very Good
**Technical Quality:** Excellent
**Clarity Of Presentation:** Excellent
**Impact:** 4

**Recommendation:**

Strong Accept: I recommend accepting the paper and will argue for my recommendation even if other reviewers hold a different opinion.

**Summary:**

The paper consider adversarial attacks against self-driving motion forecasting (prediction) models and demonstrates that adversarial training can be used to train robust prediction models. The authors highlight some challenges that are specific to training robust multi-modal prediction are discussed - namely circumventing gradient obfuscation due to stochastic futures and unstable adversarial training due to very noisy latent variables. Furthermore the authors propose balancing adversarial training with data augmentation to protect against regression on clean data. Experiments are presented to demonstrate the effectiveness of the method and highlight how the method overcomes the challenges.

**Issues:**

No major issues, just a few points discusses in the strengths and weaknesses section. Mainly the point about the realism of the threat model.

**Quality Of The Limitations Section:**

Additional details required

**Reviewer Expertise:**

4: The reviewer is confident but not absolutely certain that the evaluation is correct

**Robotics Focus:**

Highly relevant to robotics but no hardware experiments

**Strengths And Weaknesses:**

Strengths:
- The paper studies an important problem of robust motion forecasting for self-driving vehicles
- The paper also presents a robust trajectory prediction algorithm
- The authors consider performance regression on clean data and balance the adversarial training with data augmentation
- Presentation is very clear and easy to understand
- Experiments are well designed to clearly demonstrate the effectiveness of the approaches

Weaknesses
- $\delta$ has box constraints. However for a purely waypoint based parameterizations this could suffer from poor realism, i.e. zig-zagging trajectories that are kinematic ally infeasible. Therefore, some of the attacks generated can be very unrealistic and out of distribution wrt real world data, and this could be a reason why vanilla adversarial training induces a regression in performance on clean data. I am very curious about enforcing realism into the threat model (for example using a bicycle model parameterization) and seeing if there is an improvement in the performance.
- Based on my understanding, the authors propose the deterministic attack as an alternative to sampling a random latent in the inner minimization problem of the threat model. It could also be important to consider to possibility of optimizing the latent to make the attack faithful to the formulation.
- Impact on downstream planners was considered in a close-loop setting however only qualitative examples were shown.  I believe the results would be much stronger if quantitative results were presented for downstream planning.

**Summary Of Recommendation:**

The paper proposes a very important problem and motivated it  very well. Furthermore presentation was very clear, experiments were thorough, and the methodology was well formulated. The results are also very convincing.

---

### Official Review · Reviewer_zQGj · 2022-08-01

**Originality:** Fair
**Technical Quality:** Good
**Clarity Of Presentation:** Good
**Impact:** 3

**Recommendation:**

Weak Reject: I recommend rejecting the paper, but will not argue for my recommendation if the majority of other reviewers have a different opinion.

**Summary:**

The paper proposes several techniques to improve the robustness of NN-based prediction models (e.g., AgentFormer and cGAN). Specifically, the authors claim that there are several keys to defend against adversarial attacks meanwhile avoiding the performance degradation on benign data: (1) use deterministic attack to avoid graident obfuscation, (2) stablize AT with bounded noise and hybrid objective; (3) use data augmentation to enhance the performance on benign cases. Through experiments they find that their approach outperforms some existing baselines (naive AT, data augmentation, training/testing time smoothing, etc), and results in better adversarial robustness and better performance on clean data (although the clean performance is still worse than no AT).

**Issues:**

Main weakness
- The problem setting is not very practical.
- Limited technical contribution.

Others:
- The related work is too short and less helpful. Many existing adversarial approaches on self-driving are omitted.

**Quality Of The Limitations Section:**

Limitations are not well addressed

**Reviewer Expertise:**

4: The reviewer is confident but not absolutely certain that the evaluation is correct

**Robotics Focus:**

Relevant but unlikely to deploy to hardware in near future

**Strengths And Weaknesses:**

**Strengths:**

[S1] The paper is well motivated - trajectory prediction is critical to AV and there are only a few works that study the adv robustness on this problem. Figures and text were clear and easy to understand.

[S2] I like the toy examples on training with clean/adv noise on MNIST. It demonstrates the Lemma 1 and hypothesis on worse performance on clean data. The above analysis leads to the proposal of regularization term. The ablation studies on loss functions and augmentations are insightful.

[S3] The downstream evaluation on MPC-based planner is appreciated while the scenarios are still over simplified. There are only a few actors involoved in the simulator.

**Weakness:**

[W1] The adversarial attack setting is not pratical - therefore it is less interesting to the robotics community. Specifically, this paper studies the L_infty attack (0.5 or 1 m deviation is the maximum change for a standard car without shifting to another lane), this limit the actor behavior a lot and will not find interesting scenarios (lane change, more advanced interactions between agents etc) that can happen in the real world.

Moreover there are no contraints on the vehicle trajectories that are commonly used in other papers [1-3] (e.g., kinematic bicycle models, drivable area in map, constraints on acceleration, velocity, etc). Another major concern is this paper assumes an overly simplified setting - knowing the ground truth states (positions, etc) of all agents instead of taking the raw inputs (LiDAR points and camera images). This assumption on perfect perception results might lead to a large domain gap. I would suggest the authors to discuss more on the limitation section.

[1] AdvSim: Generating Safety-Critical Scenarios for Self-Driving Vehicles. Wang et al., 2021. \
[2] Generating Useful Accident-Prone Driving Scenarios via a Learned Traffic Prior. Rempe et al., 2021. \
[3] KING: Generating Safety-Critical Driving Scenarios for Robust Imitation via Kinematics Gradients. Hanselmann et al., 2022.

[W2] Limited technical novelty. The several key ingredients proposed in this paper is known in adv ML community while validated in other applications. For instance, mixing benign data and adversarial examples to avoid too much degration on clean results. Obfuscation gradients on stochastic model. Also, some previous works [1,2,3] showed that proper mixing of adv and clean examples can lead to better performance on both benign and adv examples. While in general I do appreciate the empircal validation on the prediction task, I feel a bit underwhelmed and incremental.

[1] Adversarial Examples Improve Image Recognition. Xie et al, 2020. \
[2] Adversarial Vertex Mixup: Toward Better Adversarially Robust Generalization. Lee et al, 2020. \
[3] Adversarial Attacks On Multi-Agent Communication. Tu et al, 2021.

**Others:**
- It is better to define the history state for agents explicitly (\emph{i.e.}, (x, y, t))
- Limited discussions on realted work
- The quanlitative figures are less insightful. Could we show all trajectories in a clearly way (e.g., BEV) and show the difference with no defense and other baselines.
- Fig 2 (d) font size too small
- Evaluation on PGD attack only. During the adversarial robustness evaluation on defensed models, it would be better if more attacks are tested (e.g., AutoAttack - both white and black boxes).

**Summary Of Recommendation:**

See Strengths and Weaknesses above.

---

### Official Review · Reviewer_N91T · 2022-08-05

**Originality:** Very Good
**Technical Quality:** Very Good
**Clarity Of Presentation:** Good
**Impact:** 3

**Recommendation:**

Weak Accept: I recommend accepting the paper, but will not argue for my recommendation if the majority of other reviewers have a different opinion.

**Summary:**

The main contribution of this paper is an adversarial training framework for making trajectory prediction models more robust to adversarial examples. In this setting, the adversary perturbs the trajectory of other vehicles in the scene, with the aim of reducing the accuracy of the model's predictions. This work focuses on adversarial training of conditional generative models, in particular CVAE and cGAN, which achieve state-of-the-art performance on trajectory prediction. Such models may be used in, for instance, autonomous driving systems.

There are several challenges to performing adversarial training in this setting: generating adversarial examples for stochastic models, inherent instability of adversarial training, and overfitting leading to worse performance on clean data. The proposed approach addresses all three challenges. Empirical evaluations (on the nuScenes dataset) show strong performance on adversarial inputs, while maintaining good performance on clean inputs.

**Issues:**

I would appreciate the authors' response to the main weakness I highlight above. In addition, I have a few questions about the setup.

### Questions:
* Is there a training set and a test set? Or is the adversarial training on the same dataset as it's evaluated on?
* Are the adversarial attacks ensured to be physically plausible? i.e. Are the adversarial perturbations restricted so that a car could physically follow the perturbed trajectory
* Is the adversary only allowed to change the trajectory of a single vehicle in the scene or any/all vehicles?

### Typos / unclear wording:
* Page 3, line 101: "We consider two most" → "We consider the two most"
* Page 3, footnote 1: "For simplify" → "For simplicity"
* In equations 1 and 2, shouldn't the last term be subtracted rather than added? Because a higher value of this term means higher diversity.
* In equation 2, shouldn't the first two terms be subtracted? Because we want to minimize negative log likelihood, not minimize log likelihood.
* Page 4, line 136: heading should start on a new line
* Equation 6 doesn't make sense to me. Aren't the clean loss and total (adversarial) loss two different things? Is this equation instead supposed to be a weighted sum of the two?
* Page 6, line 227: "shifting to another lanes" → "shifting to another lane"
* Page 8, Figure 4 caption: "AgenetFormer"

**Quality Of The Limitations Section:**

Limitations are addressed clearly

**Reviewer Expertise:**

3: The reviewer is fairly confident that the evaluation is correct

**Robotics Focus:**

Relevant but unlikely to deploy to hardware in near future

**Strengths And Weaknesses:**

### Strengths:
* The proposed approach is clearly explained and well-motivated, in terms of how each component addresses one of the challenges of adversarial training.
* An experiment on MNIST clearly shows how naive adversarial training can lead to degenerate learning, where the generative model just learns P(Y|Z) rather than P(Y|X, Z), in other words ignoring the input.
* The evaluation is thorough, including a comparison to a recent approach (Zhang et al.) on defending against adversarial examples in trajectory prediction.
* Includes experiments on ablations of the framework, showing how each component is necessary for obtaining the best results on both adversarial and clean inputs.

### Weaknesses:
* I'm not sure about the relevance of this work for real-world deployment. The challenge for the adversary is much more difficult in that setting, since it is no longer about just perturbing an existing trajectory in a dataset. If we imagine an adversarial driver in the scene, they would need to plan how to drive the vehicle in a certain way (which might also depend on how the other vehicles / cyclists / pedestrians in the scene will move) in order to mess up the predictions of the target vehicle.
* Some parts of the setting and experimental setup are still unclear to me - I've listed these in the Issues section below.

**Summary Of Recommendation:**

This work is well-executed and tackles an important problem of training robust trajectory prediction models. However, I am not sure about the applicability of this work to real-world robotics, and thus am hesitant to give a whole-hearted acceptance.

---

### Meta-Review · Area_Chair_sbm4 · 2022-08-14

**Recommendation:** Accept (Oral)
**Confidence:** 4

**Metareview:**

Main strengths:
1. The target problem (adversarial robustness of trajectory prediction) seems to be novel and important.
2. Experimental results are convincing for the adopted setting.

Main weaknesses:
1. The practicality of the adopted setting (e.g., assumed inputs) and threat model (e.g., actions available to adversary) need to be further justified.
2. The novelty of the methods for adversarial attack and training within the target problem should be further justified.

After the rebuttal and discussions, the reviewers have converged on acceptance. There is a sense that the work is quite interesting and novel. Though there are concerns on the practicality/generalisability of the method, the authors have sufficiently addressed the concerns.